# Fabrication of Unidirectional Water Permeable PS/PET Composite Nanofibers Modified with Silver Nanoparticles via Electrospinning

**DOI:** 10.3390/membranes13030257

**Published:** 2023-02-21

**Authors:** Chong Li, Haoyu Wang, Xiaolei Zhao, Kaihua Yang, Qinhua Meng, Longwang Zhang

**Affiliations:** 1Center for Engineering Training, Harbin Engineering University, Harbin 150001, China; 2College of Materials Science and Chemical Engineering, Harbin Engineering University, Harbin 150001, China; 3China Offshore Oil Engineering Company, Tianjin 300461, China; 4AECC Harbin Dongan Engine Co., Ltd., Harbin 150060, China

**Keywords:** electrospinning, nanofibers, water unidirectional penetration

## Abstract

In this study, the composite nanofiber membranes (AgNPs-PS/PET) composed of hydrophobic polystyrene (PS) embedded with different additions of silver nanoparticles (AgNPs) and hydrophilic hydrolyzed polyethylene terephthalate (PET) were prepared via electrospinning technology to achieve the function of unidirectional water penetration. The addition of AgNO_3_ was at 0 wt%, 0.5 wt%, 1.0 wt% and 1.5 wt% as the variables. The surface morphology and structure of AgNPs-PS/PET composite nanofibers were characterized by scanning electron microscopy (SEM), x-ray energy dispersive spectroscopy (EDS) and transmission electron microscopy (TEM). The SEM image showed that the fibers of the composite materials were continuous and uniform as a result of electrospinning. The presence and content of Ag nanoparticles dispersed in the nanofibers were investigated using EDS and TEM. The contact angle (CA) was tested to illustrate the wettability of the composite nanofiber membranes using a static contact angle measuring instrument and the process of unidirectional water penetration was recorded. Meanwhile, the mechanism of unidirectional water penetration was analyzed. Moreover, the electrospinning solution’s viscosity and conductivity were also investigated. Eventually, the optimal addition of AgNO_3_ (1.0 wt%) was confirmed and the prepared AgNPs-PS/PET composite nanofiber membranes were able to achieve the function of unidirectional water penetration. These membranes have the potential to be applied in smart textiles, unidirectional water collection and wound dressing.

## 1. Introduction

Electrospinning has become a widespread and attractive technique for producing long and continuous nanofibers with many desirable characteristics, such as the large specific area-to-volume ratio, enhanced chemical reactivity and a scalable and cost-effective fabrication process [1,2,3]. Based on the adjustable parameters of electrospinning, such as polymer type, the concentration of solutions, solvent effect and electrospinning conditions, the structures and morphologies of nanofibers can be effectively controlled; this is one of the primary advantages of electrospinning. Many polymers and inorganic/polymer composite materials, such as PAN [4], PVA [5], PLLA [6], PVP [7] and PS [8], as well as their copolymers, have been fabricated as a result of the development of electrospinning technology and applied in various fields. Among the polymeric materials, polystyrene is a widely used polymer with the advantages that it is non-toxic, tasteless, has excellent mechanical properties and good stability. Scientists have used polystyrene across diverse fields, such as tissue engineering [9], filtration [10], as an ion exchanger [11], etc. Due to its hydrophobic characteristic, many superhydrophobic PS membranes were successfully prepared by electrospinning [12,13,14]. Jiang et al. [15] obtained polystyrene nanofiber membranes by electrospinning and demonstrated that the hydrophobicity of polystyrene could be improved by electrospinning. The contact angle rose to 140°, compared to the 95° contact angle of the polystyrene films obtained by spin coating. With the excellent superhydrophobic properties of electrospun PS membranes, it is assumed that PS could be combined with some other hydrophilic materials to prepare nanofiber composite membranes to achieve unidirectional water penetration. Although there are many studies on unidirectional nanofiber membranes [16,17,18], few utilized PS to prepare the composite membranes of unidirectional water penetration via electrospinning. The PS composite nanofibers provide the possibility to prepare the materials with the unidirectional water penetrable property.

Poly(ethylene terephthalate) (PET) is one of the most important materials in both the clothing industry and in nonfibrous uses due to its excellent strength, chemical resistance, clarity, processability and dimensional stability [19,20,21,22]. Although the hydrophilicity of PET is relatively low, it is easily enhanced by hydrolysis with a strong alkali. Thus, hydrolyzed PET is usually used as hydrophilic material [23].

In addition, metal and metal oxide nanoparticles, such as AgNPs [24], CuNPs [25,26], AuNPs [27], TiO_2_ nanoparticles [28], ZnO nanoparticles [29,30], MgO nanoparticles [31] and CuO nanoparticles [31], have been successfully incorporated into electrospun membranes for uses in air filtration, for their antibacterial qualities, for photocatalysis and optical sensing, etc. By controlling the concentration of these nanoparticles in the above research, nanofiber composite membranes with small diameters and high porosity were successfully prepared, thus improving the functionality of electrospun nanofiber membranes. Among these nanoparticles, silver nanoparticles (AgNPs) are now one of the most widely used artificial nanomaterials for treating wounds and burns due to their broad-spectrum and highly effective antimicrobial properties [32,33,34]. Moreover, AgNPs have been successfully incorporated into the polymer matrices to prepare the electrospun AgNP-containing membranes with enhanced antibacterial properties [35,36]. However, few studies on incorporating AgNPs into the electrospun PS composite nanofibers have been reported. Thus, the antibacterial properties of the AgNPs-PS/PET nanofiber composite membranes might be improved based on the unidirectional water penetrable property.

In this study, AgNPs-PS/PET nanofiber composite membranes were designed and fabricated using different AgNO_3_ additions to achieve unidirectional water penetration through electrospinning. The morphology, structure and hydrophobicity of the prepared AgNPs-PS/PET nanofiber composite membranes were examined and discussed. Moreover, the viscosity and conductivity of AgNPs-PS electrospinning solutions were tested as vital influence factors in the membranes’ morphology. In addition, the property, process and mechanism of the AgNPs-PS/PET membranes’ unidirectional water penetration have also been shown and demonstrated.

## 2. Materials and Methods

### 2.1. Materials

All the chemical materials were of analytical grade. Silver nitrate (AgNO_3_, 99.8%) was purchased from Shaanxi Kaida Chemical Engineering Co., Ltd. (Shanxi, China). Sodium hydroxide (NaOH, 97%) was supplied by Tianjin Hengxing Chemical Reagent Factory (Tianjin, China). N, N-dimethyl formamide (DMF) was obtained from Tianjin Tianda Chemical Reagent Factory (Tianjin, China). Ethanoic acid (CH_3_COOH, 99.5%) and ethyl alcohol (C_2_H_6_O, 99.5%) were purchased from Jiangsu Qiangsheng Chemical Co., Ltd. (Changshu, China). Polystyrene (PS) (Mw = 104.14) was provided by Shanghai Aladdin Biochemical Technology Co., Ltd. (Shanghai, China). The PET membranes (50 g/cm^2^), with the advantage of high-temperature resistance, corrosion resistance, high air permeability and good size stability, were prepared in the laboratory via the spun-laced method.

### 2.2. Hydrolysis of PET Membranes

A rectangular sample of PET membranes was manufactured with a size of 40 cm × 4.5 cm. After heating the 20 wt% NaOH solution to 60 ℃ in the water bath, the manufactured PET membranes were soaked in the prepared solution under hydrolysis for 1 h. Then the hydrolyzed PET membranes were washed with distilled water and neutralized with 5 wt% CH_3_COOH for 5 min. To ensure that the residual reagent was thoroughly removed, the prepared film was rewashed with distilled water. Finally, the prepared PET membranes were dried and kept flat.

### 2.3. Preparation of AgNPs-PS Solution

A specific amount of PS particles were dissolved in the DMF solution. The uniform and stable PS solution (25 wt%) was obtained under magnetic stirring for 12 h. Another addition (0 wt%, 0.5 wt%, 1.0 wt%, 1.5 wt%) of AgNO_3_ was added into the PS solution and stirred by magnetic stirrer for 1 h to obtain the AgNPs-PS solution. Finally, the obtained AgNPs-PS electrospinning solution was kept in the injector for 8 h to exclude bubbles.

### 2.4. Electrospinning of AgNPs-PS/PET Composite Membranes

The electrospinning device is shown in Figure 1. With the previously obtained process parameters for PS nanofiber membranes, the injector containing the AgNPs-PS solution was connected to the spinneret with a high voltage power supply of 25 kv. The other side of the injector was fixed to a syringe pump to conduct the feed rate of 1.5 mL/h. To prepare the AgNPs-PS/PET nanofiber composite membranes, the collector roller, intertwined with PET membranes, was set at the speed of 400 r/min. Electrospinning was completed with a 12 cm work distance between the spinneret and the collection roller. During electrospinning, the solution under electric field forces overcomes the surface tension and forms the jet, that undergoes solvent evaporation and is finally deposited on the collector. After electrospinning under ultraviolet irradiation, the AgNPs-PS/PET nanofiber composite membranes, with the structure shown in Figure 2, was fabricated and then dried in the drying oven at 170 °C.

### 2.5. Material Characterizations

The morphology of AgNPs-PS nanofibers was investigated by scanning electron microscopy (SEM, HITACHI Flex1000, Tokyo, Japan) and the chemical composition of the nanofiber surface was analyzed by EDS (IXRF SDD3330, China). The average nanofiber diameter from 100 randomly drawn fibers in each SEM photo was measured by Image-J (Fiji image-J, National Institute of Health, Bethesda, MD, USA). An FEI transmission electron microscope (TEM, TECNIG20, Hillsborough, OR, USA) was used to obtain more information about the morphology of AgNO_3_-PS nanofibers. Image-J measured the average diameter of the Ag nanoparticles and the distribution of Ag particles was illustrated by TEM. The hydrophobic property of the AgNO_3_-PS/PET nanofiber composite membranes was analyzed by static contact angle measurement (OCA-20, Dataphysics, Stuttgart, Germany). The final contact angle was concluded by averaging the contact angle results of five random points of the samples. The unidirectional water penetration was measured by a camera that can record the penetration process and the duration of water dropping onto the nanofiber composite membrane. During the test, each 5 μL water drop was uniformly dropped on any of five points of the hydrophobic surface from a needle fixed 10 mm above the sample. With a shorter recorded time, the unidirectional water permeability of composite membranes was better. After leaving the dried and refurbished probe in the solution for 3 min, the conductivity of the solution was measured by the conductivity tester (DDS-11A, Shanghai Changji Geological Instrument Co., Ltd., Shanghai, China). The viscosity of the electrospinning solution was calculated from the type and rotation speed of the rotor in the rotational viscometer (NDJ79, Shanghai INESA Scientific Instrument Co., Ltd., Shanghai, China).

## 3. Results and Discussion

### 3.1. Viscosity and Conductivity of Electrospinning Solution

The viscosity and conductivity of the electrospinning solution is one of the vital factors influencing the morphology and structure of nanofibers under certain electrospinning processing parameters [37,38,39]. Thus, it is essential to further test the viscosity and conductivity of the electrospinning solution and analyze the possible influence on the morphology of nanofibers.

The viscosity of the electrospinning solution with different additions of AgNO_3_ is shown in Figure 3. With the addition of AgNO_3_ from 0 wt% to 0.5 wt%, the viscosity of the electrospinning solution rapidly decreased from 674 to 585 mPa·s. Under the salt sensitivity effect of polymer, the long-chain molecule of polymer (Rg) with the addition of metal cation was easily curled and aggregated to reduce its own entanglement, leading to the decreased viscosity of the polymer solution, as shown in Figure 4. When the addition of AgNO_3_ increased to the range of 0.5 wt% to 1.5 wt%, the viscosity of the electrospinning solution decreased slowly and tended to stabilize because of the salt sensitivity of the polymer. However, with excessive addition of metal cation, the electrospinning solution would flocculate to form sediments. Thus, the addition of AgNO_3_ had a significant effect on the viscosity. However, during the electrospinning process, it was difficult to form the continuous nanofibers with an electrospinning solution with a lower viscosity due to the decreased entanglement of molecules. Conversely, it was difficult to stretch (as a result of coulomb forces) the electrospinning solution with a higher viscosity and hyperentangled molecular chains, but easier to form the uneven fibers due to the resistance of electrospinning. The appropriate viscosity of the electrospinning solution was the key to generating a continuous fiber. This was achieved by controlling the addition of AgNO_3_ during the electrospinning process.

As shown in Figure 5, due to the intrinsic conductivity (5.02 μS/cm) of DMF, the conductivity of the electrospinning solution increased from 5.02 to 102.8 μS/cm with the addition of AgNO_3_ from 0 wt% to 1.5 wt%. The lower conductivity of the electrospinning solution resulted in a low charge density in the jet, and therefore the jets in the electric field were not subjected to sufficient stretching force, resulting in a bead-like structure rather than continuous nanofibers [40]. Moreover, the electrospinning solution with higher conductivity accelerated the jet speed and shortened the electrospinning time under the larger electric force. This made it difficult to form a Taylor cone and to generate the continuous nanofibers. Thus, the appropriate conductivity of the electrospinning solution was a guarantee of generating continuous nanofibers by controlling the addition of AgNO_3_ during the electrospinning process.

### 3.2. Surface Morphology of AgNPs-PS Side in the Composite Membranes

Figure 6 shows the SEM of AgNPs-PS/PET composite nanofibers (AgNPs-PS side) with different AgNO_3_ additions. The corresponding diameters of the nanofibers is listed in Table 1. All the nanofibers were fine and randomly arranged with the average diameter’s range of 2.08–3.38 μm. With the addition of 0 wt%, a continuous nanofiber with a few beads was obtained. This continuous nanofiber features the largest average diameter (2.47 μm) due to the low electrical conductivity of the electrospinning solution. This caused the difficultly in fiber stretching during the electrospinning process. When the addition of AgNO_3_ was 1.0 wt%, the finest fiber with the smallest average diameter of 2.08 μm was obtained, corresponding to the most uniform diameter distribution. Although the nanofibers without beads were continuous at the loading of 0.5 wt% and 1.5 wt%, the uniformity of the nanofibers decreased as the broader average diameter increased. In particular, during the addition of 1.5 wt%, the distribution of diameter became uneven, and the diameter reached its maximum (2.89 μm) due to the decreased viscosity of the electrospinning solution and the large stretching force generated by the corresponding higher conductivity of the solution in the electric field. Although the viscosity of the electrospinning solution could maintain the formation of the nanofibers, the stretching of the jet in the electric field was unstable.

As can be seen, the fine and uniform nanofiber without beads was obviously obtained with the addition of AgNO_3_, illustrating that the jet with lower viscosity was sufficiently stretched under the larger electric force generated by more charge density in the electric field. Meanwhile, the presence of AgNO_3_ increased the thermal conductivity of the jet and further accelerated the evaporation of the solvent i the jet in the electrostatic field, resulting in the generation of the fine nanofiber. However, with the addition of AgNO_3_ above 1.0 wt%, the diameter of the nanofibers became wider, and the uniformity of the nanofibers decreased. Combined with the viscosity and conductivity curve of the electrospinning solution (in Figure 3 and Figure 5), the viscosity decrease rate of the electrospinning solution was slower than the conductivity increase rate. Therefore, with the larger electric force of the jet and the shorter duration of jet flow, the jet stretching in the electric field began to be unstable with the wider nanofiber diameter despite the nanofiber still forming under the function of viscosity. It is also found that conductivity played the primary role in influencing nanofiber formation with the AgNO_3_ addition of 0–1.0 wt%, and that viscosity played the primary role with AgNO_3_ addition above 1.0 wt%. The variance and the average diameter showed that the most uniform fibers with the smallest diameter were obtained at the AgNO_3_ addition of 1.0 wt% under the most stable electrospinning. The 1.0 wt% was the optimum AgNO_3_ addition for the preparation of AgNPs-PS/PET nanofiber composite membranes.

### 3.3. Composition of AgNPs-PS Side in the Composite Membrane

Figure 7b and Table 2 show the elemental composition in the surface of AgNPs-PS nanofibers (1.0%addition of AgNO_3_) by SEM-EDS. It exhibited that no nitrogen existed in the surface with the 1.151% Ag. In some studies [41], it is mentioned that Ag^+^ can be reduced to AgNPs in DMF solvent due to the long reduction time (>24 h). However, the pretreatment time in this experiment was shorter (8 h). Therefore, we believe that more Ag^+^ can be reduced to AgNPs by the electrons provided by the high intensity electric field during the electrospinning process, and that the generated AgNPs can be uniformly dispersed on the surface or inside the nanofibers [34,42,43]. Considering this, the disappearance of nitrogen could indicate that the AgNO_3_ in the solution was reduced to Ag under electrochemical reaction. The smooth surface of the nanofibers was easily seen in Figure 7a, but it is difficult to directly observe the obvious AgNPs or zoning phenomenon by SEM due to the small volume of Ag. Therefore, the morphology and distribution of Ag in the nanofibers were investigated through the deeper observation by TEM.

Figure 8a shows that the structure of AgNPs-PS nanofiber was relatively uniform and continuous without evident holes or cracks by TEM. From the top and thin part of the nanofiber, the black round particles inferred as AgNPs were observed inside the nanofiber combined with result of EDS in Figure 8b. Moreover, the average diameter (5.46 nm) and distribution (2.81 × 10^2^ unit/μm^2^) of AgNPs was respectively calculated by Image J. It further illustrated that the AgNPs could be fabricated and uniformly dispersed in PS fibers without metal particle agglomeration by electrospinning.

### 3.4. Contact Angle and Average Time of Penetration

Figure 9 and Table 3 show the contact angle and average penetration time of AgNPs-PS/PET nanofiber membranes with various additions of AgNO_3_. According to the AgNPs-PS side of the composite membranes, an increase in the contact angle was observed by incorporation of Ag^+^ in the electrospinning solutions. Madaeni and Akbarzadeh Arbatan [44] reported the enhancement of the contact angle and hydrophobicity of nanofibers as a result of the presence of AgNPs. The contact angle increased from 124.56° to 130.49° and then decreased to 128.69° as the addition of AgNO_3_ increased. All the contact angles of the nanofiber composites were higher than 90°, showing that the AgNPs-PS layer was hydrophobic. However, the change in contact angle was relevant to the morphology of the nanofiber composites with the addition of AgNO_3_ [45]: if the addition of AgNO_3_ was too low or too high, the morphology without the uniform nanofiber was so poor that the surface energy was too high to obtain the lower contact angle. With the addition of 0.5 wt% and 1.0 wt%, the morphology of the nanofiber was better with the continuous and uniform nanofiber, and the contact angle was higher to reflect the increased hydrophobicity due to the lower surface energy.

Based on the hydrophobic-hydrophilic structure of the composite membranes, the water droplet could penetrate from the hydrophobic AgNPs-PS side to the hydrophilic PET side and couldn’t penetrate when the membrane was flipped. In addition, the shorter the penetration time, the better the unidirectional penetration property. From the results of the penetration time, the function of unidirectional liquid spreading of the AgNPs-PS/PET nanofiber membrane was achieved. According to the average penetration time, the unidirectional liquid spreading property with the addition of 0.5 wt% and 1.0 wt% was better than the other additions. Considering other factors, including morphology, the optimum addition of AgNO_3_ (1.0 wt%) was obtained for the preparation of the AgNPs-PS/PET nanofiber membrane.

### 3.5. Mechanism of Unidirectional Water Penetration

The AgNPs-PS/PET nanofiber composite membranes were considered as those membranes consisting of the hydrophobic AgNPs-PS layer and the hydrophilic PET layer. Figure 10 shows that the two layers could be combined with an obvious interface at the addition of 1.0 wt%. The process of the drop passing through the membrane from the hydrophobic AgNPs-PS layer was recorded by the video in Figure 11a, as well as the process of the drop passing through the membrane from the hydrophilic PET layer in Figure 11b. The process shows that the unidirectional function was successfully achieved with the porous structure of the AgNPs-PS layer and the moisture gradient between two layers. The mechanism can be explained as follows:

When a water droplet met the hydrophobic surface (Figure 11a), it experienced hydrostatic pressure and hydrophobic force to repel through the pores. The water drop remained in the Wenzel–Cassie state for a short time and couldn’t penetrate the layer [46]. However, as time passed, the liquid level became tilted under the action of gravity and generated the Laplace force under surface tension. Gravity and the Laplace force caused the water to penetrate the layer and the penetration depth of the water increased. Finally, the water droplet met the hydrophilic PET layers and was drawn towards the PET by the capillary forces. The droplet then began to penetrate the hydrophobic surface and was absorbed by the PET layer. On the other hand, when the drop met the PET layer, the water quickly spread to form a thin film of water and was then absorbed by the PET. However, it was unable to penetrate the AgNPs-PS layer due to the hydrophobic force acting on it. Thus, the unidirectional liquid transport from the hydrophobic AgNPs-PS layer to the hydrophilic PET layer was successfully achieved.

## 4. Conclusions

In conclusion, hybrid nanofiber membranes exhibiting the function of unidirectional water penetration were successfully prepared with the composite of PS-PET containing AgNPs produced via electrospinning. According to the AgNPs-PS side of the composite membranes, the morphology of the obtained composites using all the additions of AgNO_3_ was relatively smooth, marked by continuous and uniform nanofibers. The finest nanofiber and the optimum average diameter were obtained at the AgNO_3_ loading of 1.0 wt%, caused by the effect of the addition of AgNO_3_ on the viscosity and electrical conductivity of electrospinning solutions. Through EDS and TEM results, it was demonstrated that Ag nanoparticles were successfully introduced into the nanofiber composites. By designing the structure of the composite membranes and testing the contact angle and the penetration time, the AgNPs-PS/PET composites could be considered as hydrophobic. The hydrophobic AgNPs-PS membrane with a contact angle >90° was bonded to the hydrophilic PET membranes. The result was that the function of unidirectional penetration of water was achieved by the hydrophobic-hydrophilic composite structure (Figure 2). The mechanism of unidirectional penetration was accordingly discussed. Furthermore, the prepared composites are potential candidates for wound dressing and in self-cleaning device applications. In addition, the applications of as-prepared AgNPs-PS/PET nanofiber composite membranes with the property of unidirectional water penetration in antibacterial processes will be further investigated.

## Figures and Tables

**Figure 1 membranes-13-00257-f001:**
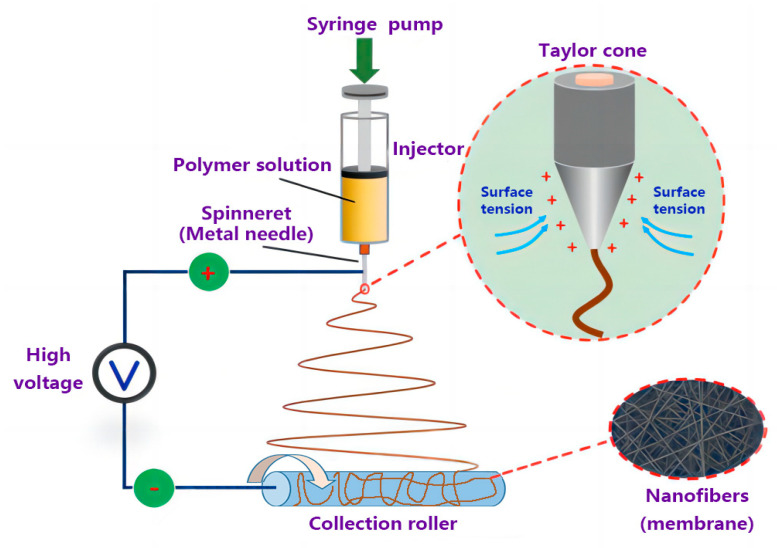
The electrospinning device.

**Figure 2 membranes-13-00257-f002:**
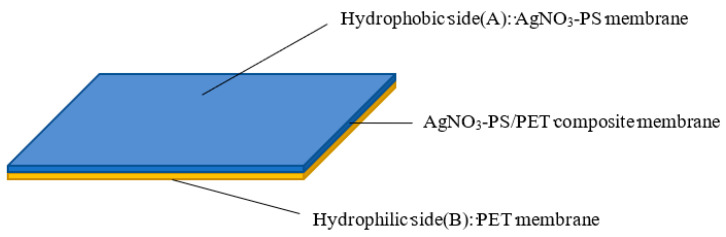
Structural map of AgNO_3_-PS/PET composite membrane.

**Figure 3 membranes-13-00257-f003:**
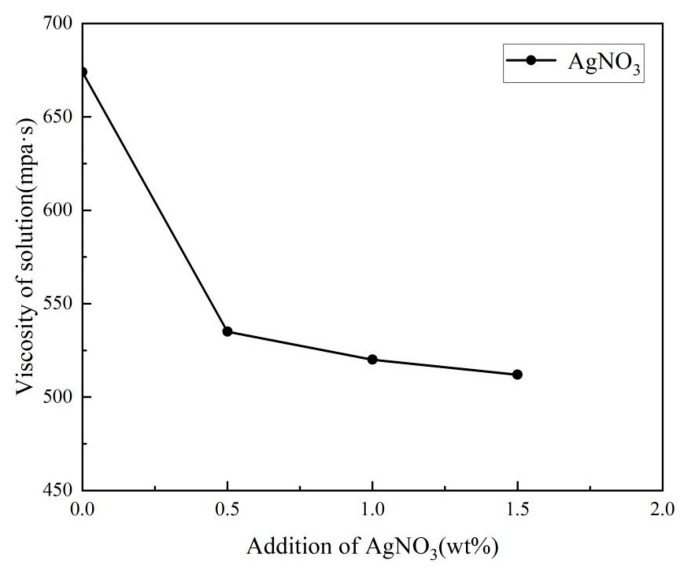
The viscosity of electrospinning solution with different addition of AgNO_3_.

**Figure 4 membranes-13-00257-f004:**
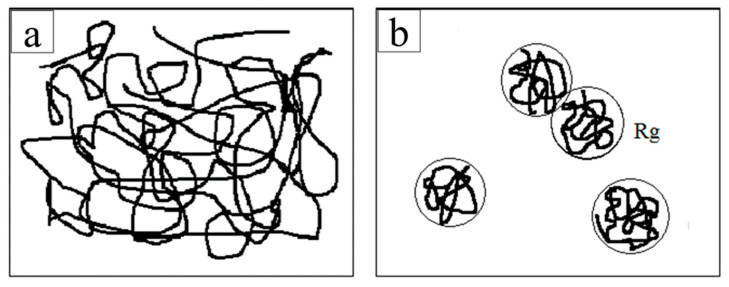
The entangled state of the polymer in the solution: (**a**) polymer solution without metal cation; (**b**) polymer solution with metal cation.

**Figure 5 membranes-13-00257-f005:**
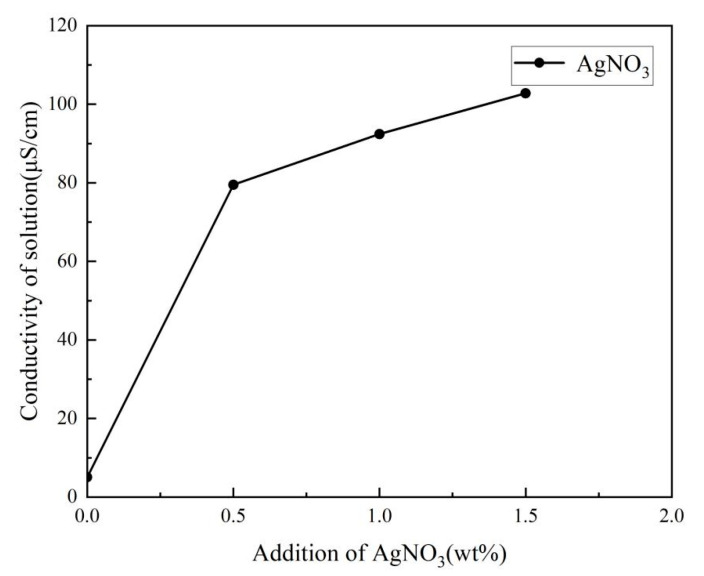
The conductivity of electrospinning solution with different addition of AgNO_3_.

**Figure 6 membranes-13-00257-f006:**
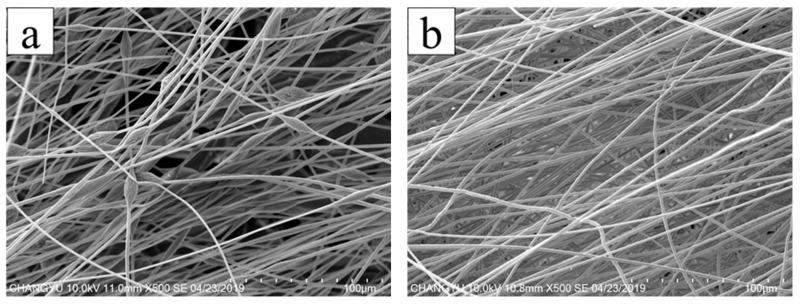
SEM image of the AgNPs-PS/PET composite membranes with different addition of AgNO_3_: (**a**) 0 wt%; (**b**) 0.5 wt%; (**c**) 1.0 wt%; (**d**) 1.5 wt%.

**Figure 7 membranes-13-00257-f007:**
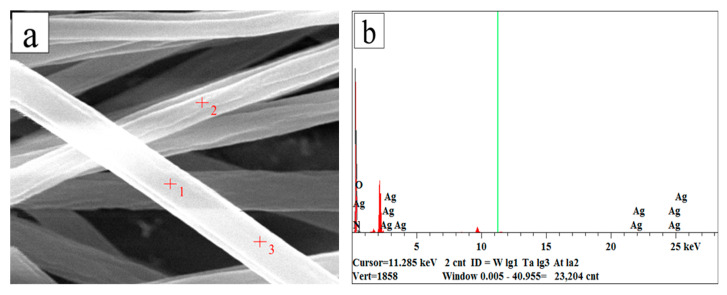
SEM and EDS of AgNPs-PS side in the composite membrane: (**a**) SEM image of AgNPs-PS nanofiber with the chosen position; (**b**) EDS image of AgNPs-PS nanofiber.

**Figure 8 membranes-13-00257-f008:**
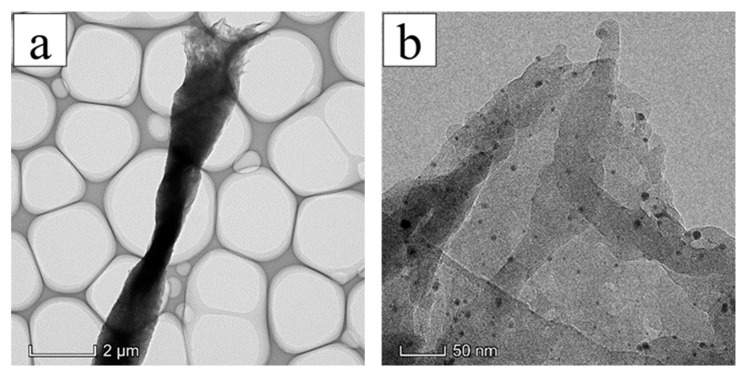
TEM image of AgNPs-PS side in the composite membrane: (**a**) TEM morphology of nanofibers at low power; (**b**) TEM morphology of nanofibers at high power.

**Figure 9 membranes-13-00257-f009:**
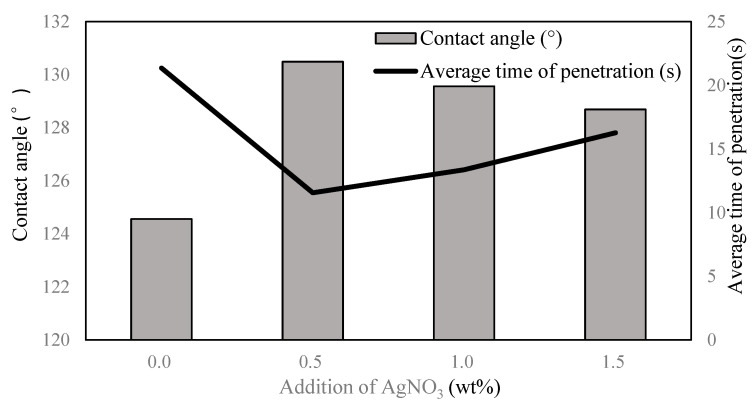
Contact angle and average penetration time of the AgNPs-PS/PET nanofiber membrane at different addition of AgNO_3_.

**Figure 10 membranes-13-00257-f010:**
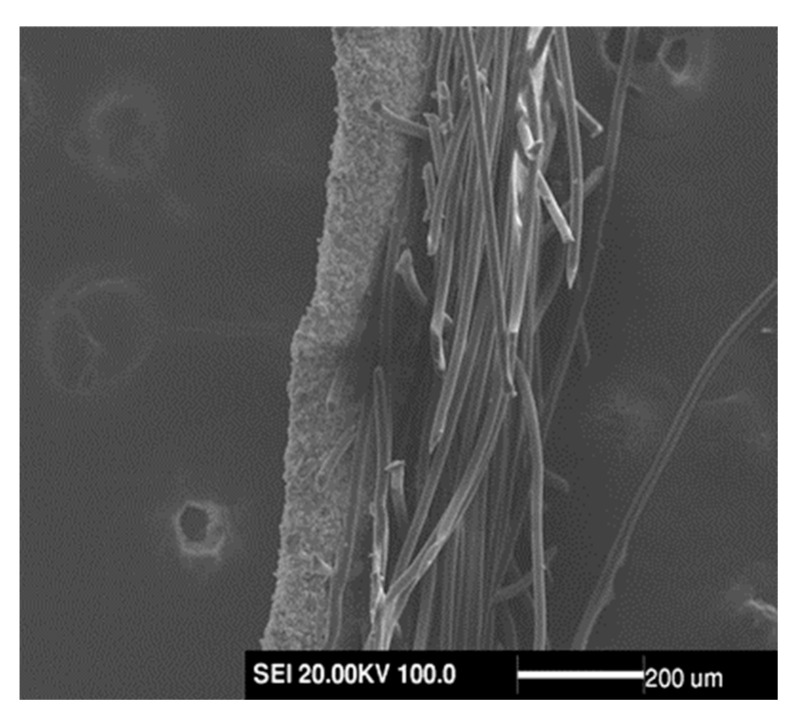
Interface SEM image between hydrophilic layer and hydrophobic layer.

**Figure 11 membranes-13-00257-f011:**
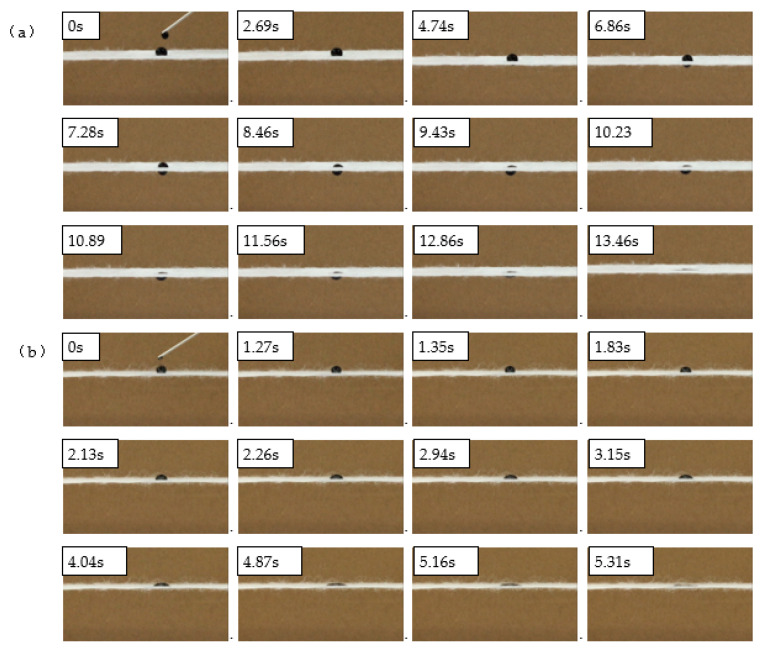
Practical effect of the unidirectional liquid transportation of AgNO_3_-PS/PET membrane: (**a**) the transportation of the drop on the hydrophobic AgNO_3_-PS layer; (**b**) the transportation of the drop on the hydrophilic PET layer.

**Table 1 membranes-13-00257-t001:** Effect of nanofiber diameter with different addition of AgNO_3_.

NO.	AgNO_3_ Concentration (wt%)	Average Diameter (μm)	Variance	Minimum Diameter (μm)	Maximum DIAMETER (μm)
a	0	2.82	2.47	2.79	7.68
b	0.5	3.38	1.23	2.27	6.42
c	1	2.08	0.77	1.02	4.04
d	1.5	2.89	2.05	2.32	9.29

**Table 2 membranes-13-00257-t002:** Content of various elements in AgNPs-PS nanofiber membrane.

Element Type	Element Content %	Unit	Error
C	94.644	wt%	1.988
N	0.000	wt%	0.000
O	4.205	wt%	3.513
Ag	1.151	wt%	1.310

**Table 3 membranes-13-00257-t003:** Effect on the unidirectional liquid transport of AgNPs-PS/PET nanofiber membrane with different addition of AgNO_3._

Addition of AgNO_3_ (wt%)	Contact Angle (°)	Average Time of Penetration (s)
0	124.56	21.36
0.5	130.49	11.56
1.0	129.56	13.36
1.5	128.69	16.27

## Data Availability

The data presented in this study are available on request from the corresponding author. The data are not publicly available as the data is a part of an ongoing study.

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
