# Peer review of "Fabrication of Unidirectional Water Permeable PS/PET Composite Nanofibers Modified with Silver Nanoparticles via Electrospinning"

_membranes, 2023, doi:10.3390/membranes13030257_

Round 1
Reviewer 1 Report
The topic is very interesting. The paper contains some minor editing and grammatical mistakes. Please revise it before publication. Line 331 - there is no patent listed. Authers may forget to include, or they could delete this section.
I recommend publishing this paper after minor revision for editing and grammatical revision.
Author Response
Dear reviewer:
Thanks for your review.For your comments, we have modified the paper and reply to your each opinion with the attached files. And we are looking forward to your check.
Wish you have a good day!
Yours sincerely,
the author

Reviewer 2 Report
Even though this work is very interesting, but it is common to see this type of membrane which made from PS/PET composited with silver nitrate, what would be the final application of this membrane. the authors mentioned silver nitrate exhibited great antimicrobial properties, but it is not their results or not compared to the literature. The characterization is very poor, it lacks interpretation and discussion with relevant references, therefore its conclusions are not they are strong.
Author Response

(The authors gave the same response as above.)

Reviewer 3 Report
There is no mention of PET in the first parts of the introduction. It is only mentioned in the last paragraph of the introduction without any justification.
I cannot see the reason behind showing the electrospinning setup in figure 1 (which is a very common setup and has been shown many times before) and also the concept of measuring contact angle in figure 3.
Figure 1 does not show any particular specific innovation/modification in the electrospinning setup (or at least I am not able to see it).
Moreover, there is no need to show the contact angle measurement in figure 3 when there the manuscript has nothing to do with the optimization/improvement/or furthering the know-how of this method.
I don’t understand why the authors dedicated two of their main figures to something “basic” and “obvious.”
The authors should have put more effort into illustrating the concept of “unidirectional electrospinning” instead of diving into non-essentials.
Author Response

(The authors gave the same response as above.)

Reviewer 4 Report
The manuscript, “Fabrication of Unidirectional Water Permeable PS/PET Composite Nanofibers Modified with Silver Particles via Electrospinning,” demonstrates a method in which the authors have designed and fabricated a nanofiber composite membrane by varying different amounts of AgNO3 to achieve the function of the unidirectional penetration of water. The authors also discussed the effect of viscosity and conductivity on the morphology of the nanofibers and the mechanism of unidirectional penetration. The study is reasonably designed and conducted, but the work cannot be publishable in its current form. So I recommend it for publication only after the minor revision, and also the authors have to address the important issues/comments mentioned below;
1. In the title, it's better to use ‘Silver Nanoparticles” instead of Silver Particles.
2. In section 3.3, the authors explained the reduction of Ag+ ions to Ag nanoparticles by the electron supplied by a high-intensity electric field during electrospinning. How are they sure it's not from the AgNO3 reduction in DMF? Check this paper: Heliyon 8 (2022) e08772 (Synthesis and characterization of polystyrene with embedded silver nanoparticle nanofibers to utilize as antibacterial and wound healing biomaterial).
3. Mention the scale bar in Figure 8 (a), it's also hard to see the Ag nanoparticles in the micrograph. What is the size of Ag nanoparticles? Check the SEM and EDS analysis in the paper mentioned in comment 2. Also, why the nanofiber diameter mentioned in the text and shown in the TEM image is different?
4. The authors should perform XRD analysis of Ag NPs embedded PS composite nanofibers.
5. The manuscript must be thoroughly spell-checked, and it should also be edited by an English-speaking native to improve the clarity of many sentences.
6. Check the heading of section 5.
7. It's better to redraw Figure 1 (electrospinning setup) To make it more visually appealing.
Author Response

(The authors gave the same response as above.)

Round 2
Reviewer 2 Report
1. In the part of the introduction, the author should add more information about the advantages feasibility, or effect of filler concentration on their diameter nanofiber or membrane due to it is directly affected to the porosity of the membrane. Several research papers have been reported these parameters please add more some relevant references, such as the effects of ZnO concentration on their diameter of nanofiber, as follows: Key Engineering Materials, 2018, 759, 81-85, doi.org/10.4028/www.scientific.net/KEM.759.81
2. Figure 3, the result of the viscosity of spinning solution versus the Ag concentration. Why the viscosity decrease with increases in the Ag concentration? As follow by Einstein's equation, the viscosity should be increased when increased the filler due to the volume fraction will be increased.
3. Looking at figure 6, why the fiber diameter of the membrane is not consistent. All of the nanofibers have been processed at the same condition?
4. Did you investigate the thermal and mechanical properties of membrane? If not, why?

Author Response

(The authors gave the same response as above.)

Reviewer 3 Report
The authors applied the proper modification to the manuscript. It is acceptable in its current format.
Author Response

(The authors gave the same response as above.)

Round 3
Reviewer 2 Report
Thank you for your revision. Please carefully check the text throughout your manuscript and considering put the more relevant references as follows: Key Engineering Materials, 2018, 759, 81-85, doi.org/10.4028/www.scientific.net/KEM.759.81